# SPIKE-BASED CAUSAL INFERENCE FOR WEIGHT ALIGNMENT

**Jordan Guerguiev[1,2], Konrad P. Kording[3], Blake A. Richards[4,5,6,7,*]**

[1] Department of Biological Sciences, University of Toronto Scarborough, Toronto, ON, Canada
[2] Department of Cell and Systems Biology, University of Toronto, Toronto, ON, Canada
[3] Department of Bioengineering, University of Pennsylvania, PA, United States
[4] Mila, Montreal, QC, Canada
[5] Department of Neurology & Neurosurgery, McGill University, Montreal, QC, Canada
[6] School of Computer Science, McGill University, Montreal, QC, Canada
[7] Canadian Institute for Advanced Research, Toronto, ON, Canada

* Corresponding author, email: blake.richards@mcgill.ca

## ABSTRACT

In artificial neural networks trained with gradient descent, the weights used for processing stimuli are also used during backward passes to calculate gradients. For the real brain to approximate gradients, gradient information would have to be propagated separately, such that one set of synaptic weights is used for processing and another set is used for backward passes. This produces the so-called "weight transport problem" for biological models of learning, where the backward weights used to calculate gradients need to mirror the forward weights used to process stimuli. This weight transport problem has been considered so hard that popular proposals for biological learning assume that the backward weights are simply random, as in the feedback alignment algorithm. However, such random weights do not appear to work well for large networks. Here we show how the discontinuity introduced in a spiking system can lead to a solution to this problem. The resulting algorithm is a special case of an estimator used for causal inference in econometrics, *regression discontinuity design*. We show empirically that this algorithm rapidly makes the backward weights approximate the forward weights. As the backward weights become correct, this improves learning performance over feedback alignment on tasks such as Fashion-MNIST, SVHN, CIFAR-10 and VOC. Our results demonstrate that a simple learning rule in a spiking network can allow neurons to produce the right backward connections and thus solve the weight transport problem.

## 1 INTRODUCTION

Any learning system that makes small changes to its parameters will only improve if the changes are correlated to the gradient of the loss function. Given that people and animals can also show clear behavioral improvements on specific tasks (Shadmehr et al., 2010), however the brain determines its synaptic updates, on average, the changes in must also correlate with the gradients of some loss function related to the task (Raman et al., 2019). As such, the brain may have some way of calculating at least an estimator of gradients.

To-date, the bulk of models for how the brain may estimate gradients are framed in terms of setting up a system where there are both bottom-up, feedforward and top-down, feedback connections. The feedback connections are used for propagating activity that can be used to estimate a gradient (Williams, 1992; Lillicrap et al., 2016; Akrout et al., 2019; Roelfsema & Ooyen, 2005; Lee et al., 2015; Scellier & Bengio, 2017; Sacramento et al., 2018). In all such models, the gradient estimator is less biased the more the feedback connections mirror the feedforward weights. For example, in the REINFORCE algorithm (Williams, 1992), and related algorithms like AGREL (Roelfsema &

Ooyen, 2005), learning is optimal when the feedforward and feedback connections are perfectly symmetric, such that for any two neurons $i$ and $j$ the synaptic weight from $i$ to $j$ equals the weight from $j$ to $i$, e.g. $W_{ji} = W_{ij}$ (Figure 1). Some algorithms simply assume weight symmetry, such as Equilibrium Propagation (Scellier & Bengio, 2017). The requirement for synaptic weight symmetry is sometimes referred to as the "weight transport problem", since it seems to mandate that the values of the feedforward synaptic weights are somehow transported into the feedback weights, which is not biologically realistic (Crick, 1989-01-12; Grossberg, 1987). Solving the weight transport problem is crucial to biologically realistic gradient estimation algorithms (Lillicrap et al., 2016), and is thus an important topic of study.

Several solutions to the weight transport problem have been proposed for biological models, including hard-wired sign symmetry (Moskovitz et al., 2018), random fixed feedback weights (Lillicrap et al., 2016), and learning to make the feedback weights symmetric (Lee et al., 2015; Sacramento et al., 2018; Akrout et al., 2019; Kolen & Pollack, 1994). Learning to make the weights symmetric is promising because it is both more biologically feasible than hard-wired sign symmetry (Moskovitz et al., 2018) and it leads to less bias in the gradient estimator (and thereby, better training results) than using fixed random feedback weights (Bartunov et al., 2018; Akrout et al., 2019). However, of the current proposals for learning weight symmetry some do not actually work well in practice (Bartunov et al., 2018) and others still rely on some biologically unrealistic assumptions, including scalar value activation functions (as opposed to all-or-none spikes) and separate error feedback pathways with one-to-one matching between processing neurons for the forward pass and error propagation neurons for the backward pass Akrout et al. (2019); Sacramento et al. (2018).

Interestingly, learning weight symmetry is implicitly a causal inference problem—the feedback weights need to represent the causal influence of the upstream neuron on its downstream partners. As such, we may look to the causal infererence literature to develop better, more biologically realistic algorithms for learning weight symmetry. In econometrics, which focuses on quasi-experiments, researchers have developed various means of estimating causality without the need to actually randomize and control the variables in question Angrist & Pischke (2008); Marinescu et al. (2018). Among such quasi-experimental methods, regression discontinuity design (RDD) is particularly promising. It uses the discontinuity introduced by a threshold to estimate causal effects. For example, RDD can be used to estimate the causal impact of getting into a particular school (which is a discontinuous, all-or-none variable) on later earning power. RDD is also potentially promising for estimating causal impact in biological neural networks, because real neurons communicate with discontinuous, all-or-none spikes. Indeed, it has been shown that the RDD approach can produce unbiased estimators of causal effects in a system of spiking neurons Lansdell & Kording (2019). Given that learning weight symmetry is fundamentally a causal estimation problem, we hypothesized that RDD could be used to solve the weight transport problem in biologically realistic, spiking neural networks.

Here, we present a learning rule for feedback synaptic weights that is a special case of the RDD algorithm previously developed for spiking neural networks (Lansdell & Kording, 2019). Our algorithm takes advantage of a neuron's spiking discontinuity to infer the causal effect of its spiking on the activity of downstream neurons. Since this causal effect is proportional to the feedforward synaptic weight between the two neurons, by estimating it, feedback synapses can align their weights to be symmetric with the reciprocal feedforward weights, thereby overcoming the weight transport problem. We demonstrate that this leads to the reduction of a cost function which measures the weight symmetry (or the lack thereof), that it can lead to better weight symmetry in spiking neural networks than other algorithms for weight alignment (Akrout et al., 2019) and it leads to better learning in deep neural networks in comparison to the use of fixed feedback weights (Lillicrap et al., 2016). Altogether, these results demonstrate a novel algorithm for solving the weight transport problem that takes advantage of discontinuous spiking, and which could be used in future models of biologically plausible gradient estimation.

## 2   RELATED WORK

Previous work has shown that even when feedback weights in a neural network are initialized randomly and remain fixed throughout training, the feedforward weights learn to partially align themselves to the feedback weights, an algorithm known as *feedback alignment* (Lillicrap et al., 2016).

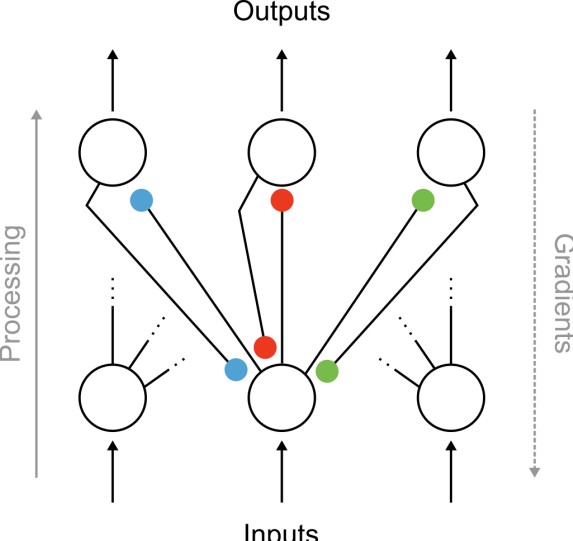

Figure 1: Illustration of weight symmetry in a neural network with feedforward and feedback connections. Processing of inputs to outputs is mirrored by backward flow of gradient information. Gradient estimation is best when feedback synapses have symmetric weights to feedforward synapses (illustrated with colored circles).

While feedback alignment is successful at matching the learning performance of true gradient descent in relatively shallow networks, it does not scale well to deeper networks and performs poorly on difficult computer vision tasks (Bartunov et al., 2018).

The gap in learning performance between feedback alignment and gradient descent can be overcome if feedback weights are continually updated to match the sign of the reciprocal feedforward weights (Moskovitz et al., 2018). Furthermore, learning the feedback weights in order to make them more symmetric to the feedforward weights has been shown to improve learning over feedback alignment (Akrout et al., 2019).

To understand the underlying dynamics of learning weight symmetry, Kunin et al. (2019) define the *symmetric alignment* cost function, $\mathcal{R}_{\text{SA}}$, as one possible cost function that, when minimized, leads to weight symmetry:

$$
\begin{aligned}
\mathcal{R}_{\text{SA}} &:= \|W - Y^T\|_F^2 \\
&= \|W\|_F^2 + \|Y\|_F^2 - 2\text{tr}(WY)
\end{aligned}
\tag{1}
$$

where $W$ are feedforward weights and $Y$ are feedback weights. The first two terms are simply weight regularization terms that can be minimized using techniques like weight decay. But, the third term is the critical one for ensuring weight alignment.

In this paper we present a biologically plausible method of minimizing the third term. This method is based on the work of Lansdell & Kording (2019), who demonstrated that neurons can estimate their causal effect on a global reward signal using the discontinuity introduced by spiking. This is accomplished using RDD, wherein a piecewise linear model is fit around a discontinuity, and the differences in the regression intercepts indicates the causal impact of the discontinuous variable. In Lansdell & Kording (2019), neurons learn a piece-wise linear model of a reward signal as a function of their input drive, and estimate the causal effect of spiking by looking at the discontinuity at the spike threshold. Here, we modify this technique to perform causal inference on the effect of spiking on downstream neurons, rather than a reward signal. We leverage this to develop a learning rule for feedback weights that induces weight symmetry and improves training.

## 3 OUR CONTRIBUTIONS

The primary contributions of this paper are as follows:

- We demonstrate that spiking neurons can accurately estimate the causal effect of their spiking on downstream neurons by using a piece-wise linear model of the feedback as a function of the input drive to the neuron.

- We present a learning rule for feedback weights that uses the causal effect estimator to encourage weight symmetry. We show that when feedback weights update using this algorithm it minimizes the symmetric alignment cost function, $\mathcal{R}_{SA}$.

- We demonstrate that this learning weight symmetry rule improves training and test accuracy over feedback alignment, approaching gradient-descent-level performance on Fashion-MNIST, SVHN, CIFAR-10 and VOC in deeper networks.

## 4 METHODS

### 4.1 GENERAL APPROACH

In this work, we utilize a spiking neural network model for aligning feedforward and feedback weights. However, due to the intense computational demands of spiking neural networks, we only use spikes for the RDD algorithm. We then use the feedback weights learned by the RDD algorithm for training a non-spiking convolutional neural network. We do this because the goal of our work here is to develop an algorithm for aligning feedback weights in spiking networks, not for training feedforward weights in spiking networks on other tasks. Hence, in the interest of computational expediency, we only used spiking neurons when learning to align the weights. Additional details on this procedure are given below.

### 4.2 RDD FEEDBACK TRAINING PHASE

At the start of every training epoch of a convolutional neural network, we use an RDD feedback weight training phase, during which all fully-connected sets of feedback weights in the network are updated. To perform these updates, we simulate a separate network of leaky integrate-and-fire (LIF) neurons. LIF neurons incorporate key elements of real neurons such as voltages, spiking thresholds and refractory periods. Each epoch, we begin by training the feedback weights in the LIF network. These weights are then transferred to the convolutional network, which is used for training the feedforward weights. The new feedforward weights are then transferred to the LIF net, and another feedback training phase with the LIF net starts the next epoch (Figure 2A). During the feedback training phase, the LIF network undergoes a training phase lasting 90 s of simulated time (30 s per set of feedback weights) (Figure 2B). We find that the spiking network used for RDD feedback training and the convolutional neural network are very closely matched in the activity of the units (Figure S1), which gives us confidence that this approach of using a separate non-spiking network for training the feedforward weights is legitimate.

During the feedback training phase, a small subset of neurons in the first layer receive driving input that causes them to spike, while other neurons in this layer receive no input (see Appendix A.2). The subset of neurons that receive driving input is randomly selected every 100 ms of simulated time. This continues for 30 s in simulated time, after which the same process occurs for the subsequent hidden layers in the network. This protocol enforces sparse, de-correlated firing patterns that improve the causal inference procedure of RDD.

### 4.3 LIF DYNAMICS

During the RDD feedback training phase, each unit in the network is simulated as a leaky integrate-and-fire neuron. Spiking inputs from the previous layer arrive at feedforward synapses, where they are convolved with a temporal exponential kernel to simulate post-synaptic spike responses $\mathbf{p} = [p_1, p_2, ..., p_m]$ (see Appendix A.1). The neurons can also receive driving input $\tilde{p}_i$, instead of synaptic inputs. The total feedforward input to neuron $i$ is thus defined as:

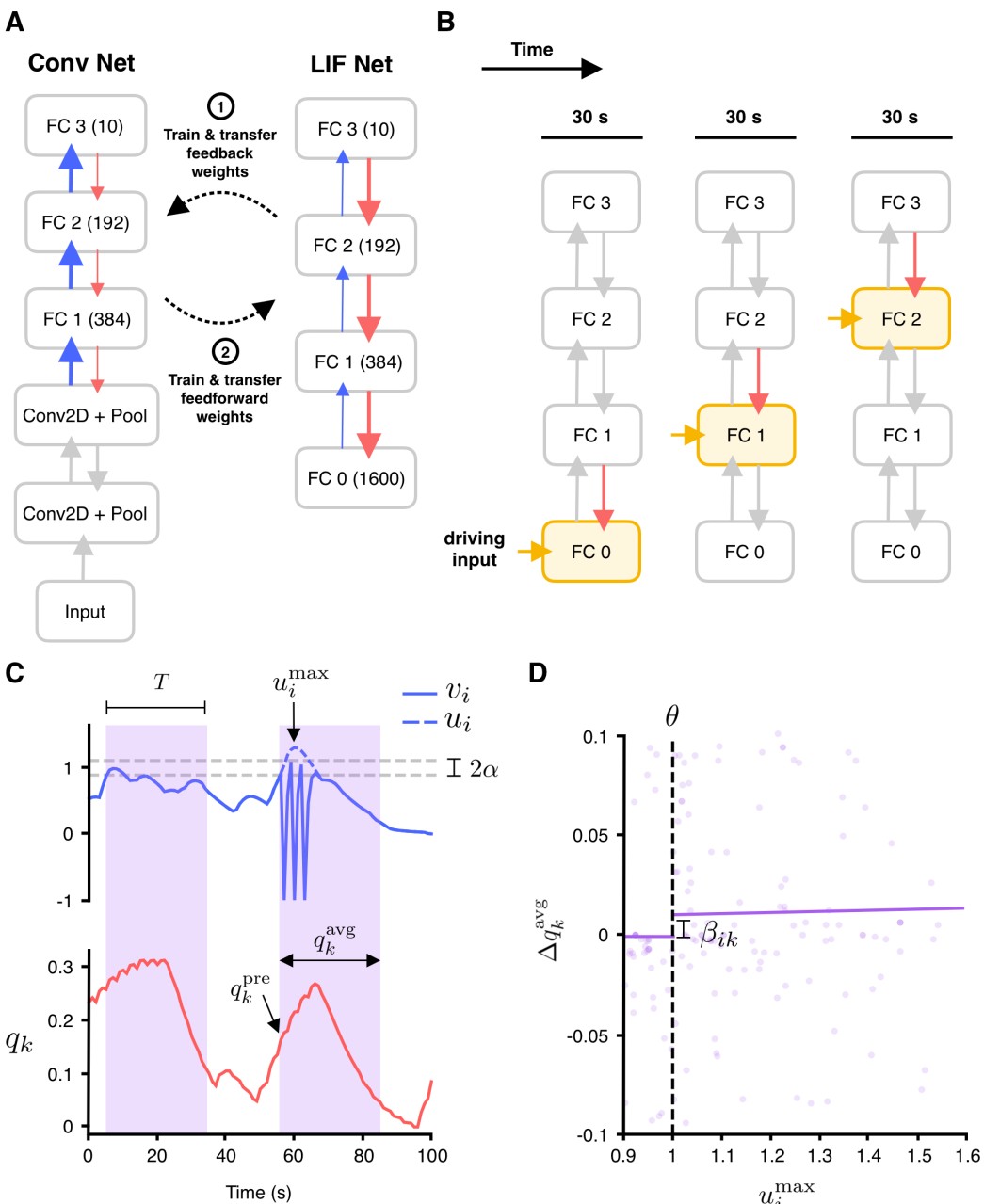

Figure 2: **A.** Layers of the convolutional network trained on CIFAR-10 and the corresponding network of LIF neurons that undergoes RDD feedback training. Fully-connected feedforward weights (blue) and feedback weights (red) are shared between the two networks. Every training epoch consists of an RDD feedback training phase where feedback weights in the LIF net are updated (bold red arrows) and transferred to the convolutional net, and a regular training phase where feedforward weights in the convolutional net are updated (bold blue arrows) and transferred back to the LIF net. **B.** RDD feedback training protocol. Every 30 s, a different layer in the LIF network receives driving input and updates its feedback weights (red) using the RDD algorithm. **C.** *Top:* Sample voltage ($v_i$, solid line) and input drive ($u_i$, dashed line) traces. Whenever $v_i$ approaches the spiking threshold, an RDD window lasting $T$ ms is triggered. $u_i^{\mathrm{max}}$ is the maximum input drive during this window of time. *Bottom:* Feedback received at a synapse, $q_k$. $q_k^{\mathrm{pre}}$ is the feedback signal at the start of an RDD window, while $q_k^{\mathrm{avg}}$ is the average of the feedback signal during the time window. **D.** Samples of $\Delta q_k^{\mathrm{avg}}$ vs. $u_i^{\mathrm{max}}$ are used to update a piece-wise linear function of $u_i^{\mathrm{max}}$, and the causal effect $\beta_{ik}$ is defined as the difference of the left and right limits of the function at the spiking threshold.

$$I_i := \begin{cases} \omega \tilde{p}_i & \text{if } \tilde{p}_i > 0 \\ \sum_{j=1}^{m} W_{ij} p_j & \text{otherwise} \end{cases} \tag{2}$$

where $W_{ij}$ is the feedforward weight to neuron $i$ from neuron $j$ in the previous layer, and $\omega$ is a hyperparameter. The voltage of the neuron, $v_i$, evolves as:

$$\frac{dv_i}{dt} = -g_L v_i + g_D (I_i - v_i) \tag{3}$$

where $g_L$ and $g_D$ are leak and dendritic conductance constants, respectively. The *input drive* to the neuron, $u_i$, is similarly modeled:

$$\frac{du_i}{dt} = -g_L u_i + g_D (I_i - u_i) \tag{4}$$

If the voltage $v_i$ passes a spiking threshold $\theta$, the neuron spikes and the voltage is reset to a value $v_{\text{reset}} = -1$ (Figure 2C). Note that the input drive does not reset. This helps us to perform regressions both above and below the spike threshold.

In addition to feedforward inputs, spiking inputs from the downstream layer arrive at feedback synapses, where they create post-synaptic spike responses $\mathbf{q} = [q_1, q_2, ..., q_n]$. These responses are used in the causal effect estimation (see below).

## 4.4 RDD ALGORITHM

Whenever the voltage approaches the threshold $\theta$ (ie. $|v_i - \theta| < \alpha$ where $\alpha$ is a constant), an RDD window is initiated, lasting $T = 30$ ms in simulated time (Figure 2C). At the end of this time window, at each feedback synapse, the maximum input drive during the RDD window, $u_i^{\text{max}}$, and the average change in feedback from downstream neuron $k$ during the RDD window, $\Delta q_k^{\text{avg}}$, are recorded. $\Delta q_k^{\text{avg}}$ is defined as the difference between the average feedback received during the RDD window, $q_k^{\text{avg}}$, and the feedback at the start of the RDD window, $q_k^{\text{pre}}$:

$$\Delta q_k^{\text{avg}} := q_k^{\text{avg}} - q_k^{\text{pre}} \tag{5}$$

Importantly, $u_i^{\text{max}}$ provides a measure of how strongly neuron $i$ was driven by its inputs (and whether or not it passed the spiking threshold $\theta$), while $\Delta q_k^{\text{avg}}$ is a measure of how the input received as feedback from neuron $k$ changed after neuron $i$ was driven close to its spiking threshold. These two values are then used to fit a piece-wise linear model of $\Delta q_k^{\text{avg}}$ as a function of $u_i^{\text{max}}$ (Figure 2D). This piece-wise linear model is defined as:

$$f_{ik}(x) := \begin{cases} c_{ik}^1 x + c_{ik}^2 & \text{if } x < \theta \\ c_{ik}^3 x + c_{ik}^4 & \text{if } x \geq \theta \end{cases} \tag{6}$$

The parameters $c_{ik}^1$, $c_{ik}^2$, $c_{ik}^3$ and $c_{ik}^4$ are updated to perform linear regression using gradient descent:

$$L = \frac{1}{2} \| f_{ik}(u_i^{\text{max}}) - \Delta q_k^{\text{avg}} \|^2 \tag{7}$$

$$\Delta c_{ik}^l \propto -\frac{\partial L}{\partial c_{ik}^l} \quad \text{for } l \in \{1, 2, 3, 4\} \tag{8}$$

An estimate of the causal effect of neuron $i$ spiking on the activity of neuron $k$, $\beta_{ik}$, is then defined as the difference in the two sides of the piece-wise linear function at the spiking threshold:

$$\beta_{ik} := \lim_{x \to \theta^+} f_{ik}(x) - \lim_{x \to \theta^-} f_{ik}(x) \tag{9}$$

Finally, the weight at the feedback synapse, $Y_{ik}$, is updated to be a scaled version of $\beta_{ik}$:

$$Y_{ik} = \beta_{ik} \frac{\gamma}{\sigma_\beta^2} \tag{10}$$

where $\gamma$ is a hyperparameter and $\sigma_\beta^2$ is the standard deviation of $\beta$ values for all feedback synapses in the layer. This ensures that the scale of the full set of feedback weights between two layers in the network remains stable during training.

## 5 RESULTS

### 5.1 ALIGNMENT OF FEEDBACK AND FEEDFORWARD WEIGHTS

To measure how well the causal effect estimate at each feedback synapse, $\beta_{ik}$, and thus the feedback weight $Y_{ik}$, reflects the reciprocal feedforward weight $W_{ki}$, we can measure the percentage of feedback weights that have the same sign as the reciprocal feedforward weights (Figure 3A). When training on CIFAR-10 with no RDD feedback training phase (ie. feedback weights remain fixed throughout training), the feedback alignment effect somewhat increases the sign alignment during training, but it is ineffective at aligning the signs of weights in earlier layers in the network. Compared to feedback alignment, the addition of an RDD feedback training phase greatly increases the sign alignment between feedback and feedforward weights for all layers in the network, especially at earlier layers. In addition, the RDD algorithm increases sign alignment throughout the hierarchy more than the current state-of-the-art algorithm for weight alignment introduced recently by Akrout et al. Akrout et al. (2019) (Figure 3A). Furthermore, RDD feedback training changes feedback weights to not only match the sign but also the magnitude of the reciprocal feedforward weights (Figure 3B), which makes it better for weight alignment than hard-wired sign symmetry (Moskovitz et al., 2018).

### 5.2 DESCENDING THE SYMMETRIC ALIGNMENT COST FUNCTION

The *symmetric alignment* cost function (Kunin et al., 2019) (Equation 1) can be broken down as:

$$\mathcal{R}_{\text{SA}} = \mathcal{R}_{\text{decay}} + \mathcal{R}_{\text{self}} \tag{11}$$

where we define $\mathcal{R}_{\text{decay}}$ and $\mathcal{R}_{\text{self}}$ as:

$$\mathcal{R}_{\text{decay}} := \|W\|_F^2 + \|Y\|_F^2 \tag{12}$$

$$\mathcal{R}_{\text{self}} := -2\text{tr}(WY) \tag{13}$$

$\mathcal{R}_{\text{decay}}$ is simply a weight regularization term that can be minimized using techniques like weight decay. $\mathcal{R}_{\text{self}}$, in contrast, measures how well aligned in direction the two matrices are. Our learning rule for feedback weights minimizes the $\mathcal{R}_{\text{self}}$ term for weights throughout the network (Figure 4). By comparison, feedback alignment decreases $\mathcal{R}_{\text{self}}$ to a smaller extent, and its ability to do so diminishes at earlier layers in the network. This helps to explain why our algorithm induces weight alignment, and can improve training performance (see below).

### 5.3 PERFORMANCE ON FASHION-MNIST, SVHN, CIFAR-10 AND VOC

We trained the same network architecture (see Appendix A.3) on the Fashion-MNIST, SVHN, CIFAR-10 and VOC datasets using standard autograd techniques (backprop), feedback alignment and our RDD feedback training phase. RDD feedback training substantially improved the network's performance over feedback alignment, and led to backprop-level accuracy on the train and test sets (Figure 5).

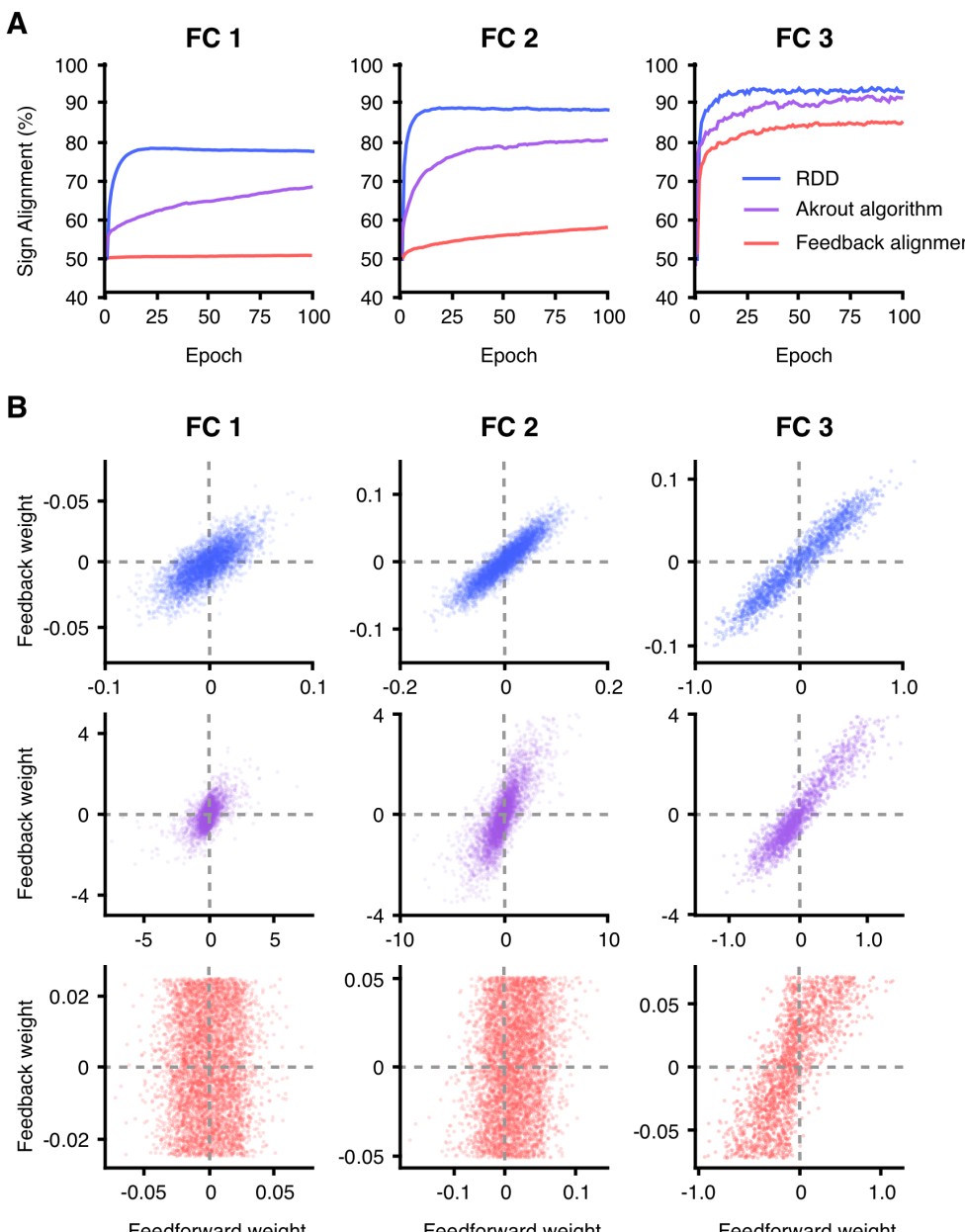

Figure 3: **A.** Evolution of sign alignment (the percent of feedforward and feedback weights that have the same sign) for each fully-connected layer in the network when trained on CIFAR-10 using RDD feedback training (blue), using the algorithm proposed by Akrout et al. (2019) (purple), and using feedback alignment (red). **B.** Feedforward vs. feedback weights for each fully-connected layer at the end of training, with RDD feedback training (blue), the Akrout algorithm (purple), and feedback alignment (red).

## 6 DISCUSSION

In order to understand how the brain learns complex tasks that require coordinated plasticity across many layers of synaptic connections, it is important to consider the weight transport problem. Here, we presented an algorithm for updating feedback weights in a network of spiking neurons that takes advantage of the spiking discontinuity to estimate the causal effect between two neurons (Figure 2). We showed that this algorithm enforces weight alignment (Figure 3), and identified a loss function, $\mathcal{R}_{\text{self}}$, that is minimized by our algorithm (Figure 4). Finally, we demonstrated that our algorithm

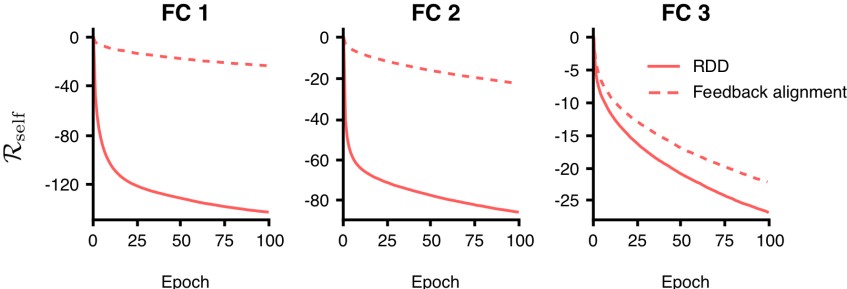

Figure 4: Evolution of $\mathcal{R}_{\mathrm{self}}$ for each fully-connected layer in the network when trained on CIFAR-10 using RDD feedback training (solid lines) and using feedback alignment (dashed lines). RDD feedback training dramatically decreases this loss compared to feedback alignment, especially in earlier layers.

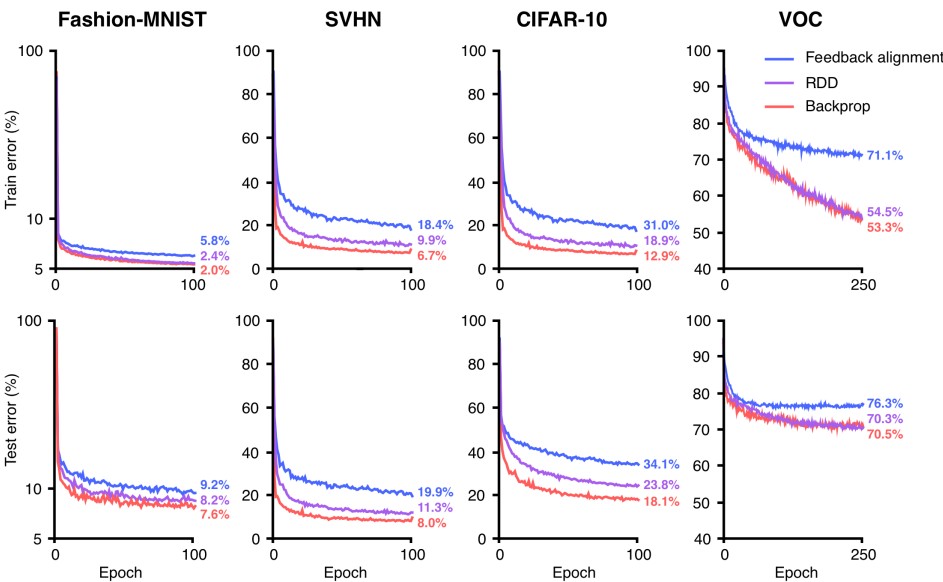

Figure 5: Comparison of Fashion-MNIST, SVHN, CIFAR-10 and VOC train error (top row) and test error (bottom row). RDD feedback training substantially improves test error performance over feedback alignment in both learning tasks.

allows deep neural networks to achieve better learning performance than feedback alignment on Fashion-MNIST and CIFAR-10 (Figure 5). These results demonstrate the potential power of RDD as a means for solving the weight transport problem in biologically plausible deep learning models.

One aspect of our algorithm that is still biologically implausible is that it does not adhere to Dale's principle, which states that a neuron performs the same action on all of its target cells (Strata & Harvey). This means that a neuron's outgoing connections cannot include both positive and negative weights. However, even under this constraint, a neuron can have an excitatory effect on one downstream target and an inhibitory effect on another, by activating intermediary inhibitory interneurons. Because our algorithm provides a causal estimate of one neuron's impact on another, theoretically, it could capture such polysynaptic effects. Therefore, this algorithm is in theory compatible with Dale's principle. Future work should test the effects of this algorithm when implemented in a network of neurons that are explicitly excitatory or inhibitory.

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

# A  APPENDIX

## A.1  LIF NEURON SIMULATION DETAILS

Post-synaptic spike responses at feedforward synapses, $\mathbf{p}$, were calculated from pre-synaptic binary spikes using an exponential kernel function $\kappa$:

$$p_j(t) = \sum_k \kappa(t - \tilde{t}_{jk}) \tag{14}$$

where $\tilde{t}_{jk}$ is the $k^{\text{th}}$ spike time of input neuron $j$ and $\kappa$ is given by:

$$\kappa(t) = (e^{-t/\tau_L} - e^{-t/\tau_S})\Theta(t)/(\tau_L - \tau_s) \tag{15}$$

where $\tau_s = 0.003$ s and $\tau_L = 0.01$ s represent short and long time constants, and $\Theta$ is the Heaviside step function. Post-synaptic spike responses at feedback synapses, $\mathbf{q}$, were computed in the same way.

## A.2  RDD FEEDBACK TRAINING IMPLEMENTATION

### A.2.1  WEIGHT SCALING

Weights were shared between the convolutional network and the network of LIF neurons, but feedforward weights in the LIF network were scaled versions of the convolutional network weights:

$$W_{ij}^{\text{LIF}} = \psi m W_{ij}^{\text{Conv}}/\sigma_{W^{\text{Conv}}}^2 \tag{16}$$

where $W^{\text{Conv}}$ is a feedforward weight matrix in the convolutional network, $W^{\text{LIF}}$ is the corresponding weight matrix in the LIF network, $m$ is the number of units in the upstream layer (ie. the number of columns in $W^{\text{Conv}}$), $\sigma_{W^{\text{Conv}}}^2$ is the standard deviation of $W^{\text{Conv}}$ and $\psi$ is a hyperparameter. This rescaling ensures that spike rates in the LIF network stay within an optimal range for the RDD algorithm to converge quickly, even if the scale of the feedforward weights in the convolutional network changes during training. This avoids situations where the scale of feedforward weights is so small that little or no spiking occurs in the LIF neurons.

### A.2.2  FEEDBACK TRAINING PARADIGM

The RDD feedback training paradigm is implemented as follows. We start by providing driving input to the first layer in the network of LIF neurons. To create this driving input, we choose a subset of 20% of the neurons in that layer, and create a unique input spike train for each of these neurons using a Poisson process with a rate of 200 Hz. All other neurons in the layer receive no driving input. Every 100 ms, a new set of neurons to receive driving input is randomly chosen. After 30 s, this layer stops receiving driving input, and the process repeats for the next layer in the network.

## A.3  NETWORK AND TRAINING DETAILS

The network architectures used to train on Fashion-MNIST and CIFAR-10 are described in Table 1.

Inputs were randomly cropped and flipped during training, and batch normalization was used at each layer. Networks were trained using a minibatch size of 32.

## A.4  AKROUT ET AL. (2019) ALGORITHM IMPLEMENTATION

In experiments that compared sign alignment using our RDD algorithm with the Akrout et al. (2019) algorithm, we kept the same RDD feedback training paradigm (ie. layers were sequentially driven, and a small subset of neurons in each layer was active at once). However, rather than updating feedback weights using RDD, we recorded the mean firing rates of the active neurons in the upstream

| Layer | Fashion-MNIST | SVHN & CIFAR-10 | VOC |
|---|---|---|---|
| Input | $28 \times 28 \times 1$ | $32 \times 32 \times 3$ | $32 \times 32 \times 3$ |
| 1 | Conv2D $5 \times 5$, 64 ReLU | Conv2D $5 \times 5$, 64 ReLU | Conv2D $5 \times 5$, 64 ReLU |
| 2 | MaxPool $2 \times 2$, stride 2 | MaxPool $2 \times 2$, stride 2 | MaxPool $2 \times 2$, stride 2 |
| 3 | Conv2D $5 \times 5$, 64 ReLU | Conv2D $5 \times 5$, 64 ReLU | Conv2D $5 \times 5$, 64 ReLU |
| 4 | MaxPool $2 \times 2$, stride 2 | MaxPool $2 \times 2$, stride 2 | MaxPool $2 \times 2$, stride 2 |
| 5 | FC 384 ReLU | FC 384 ReLU | FC 384 ReLU |
| 6 | FC 192 ReLU | FC 192 ReLU | FC 192 ReLU |
| 7 | FC 10 ReLU | FC 10 ReLU | FC 21 ReLU |

Table 1: Network architectures used to train on Fashion-MNIST, SVHN, CIFAR-10 and VOC.

layer, $\mathbf{r}^l$, and the mean firing rates in the downstream layer, $\mathbf{r}^{l+1}$. We then used the following feedback weight update rule:

$$\Delta \mathbf{Y} = \eta \mathbf{r}^l \mathbf{r}^{(l+1)^T} - \lambda_{\mathrm{WD}} \mathbf{Y} \tag{17}$$

where $Y$ are the feedback weights between layers $l + 1$ and $l$, and $\eta$ and $\lambda_{\mathrm{WD}}$ are learning rate and weight decay hyperparameters, respectively.

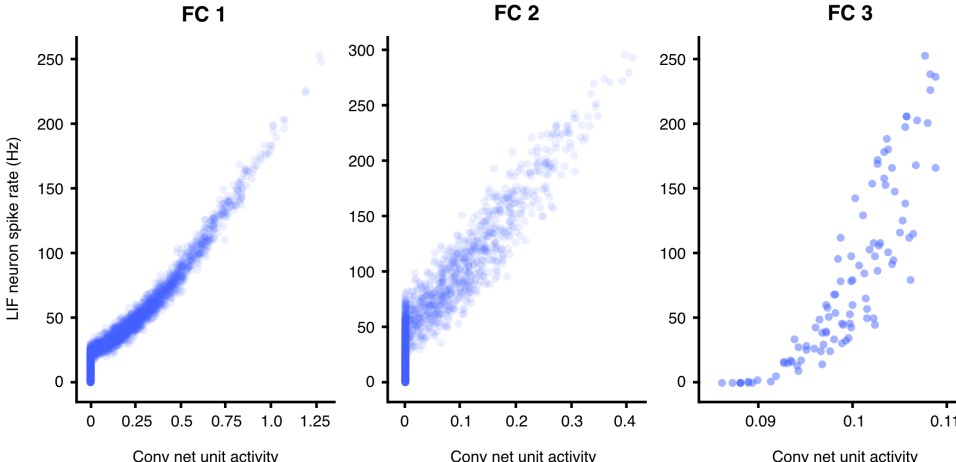

Figure S1: Comparison of average spike rates in the fully-connected layers of the LIF network vs. activities of the same layers in the convolutional network, when both sets of layers were fed the same input. Spike rates in the LIF network are largely correlated with activities of units in the convolutional network.

