# OpenReview forum: "Spike-based causal inference for weight alignment"
_ICLR.cc/2020/Conference — Accept (Poster)_

### Official Review · AnonReviewer2 · 2019-10-15
**Official Blind Review #2**

**Rating:** 6

**Review:**

The paper introduces a training mechanism for spiking neural nets that employs a causal inference technique, called RDD, for adjustment of backward spiking weights. This technique induces the backward influence strengths to be reciprocal to the forward ones, bringing desirable symmetry properties.

Pros:
 * The relationship between causal inference and biologically plausible learning is very interesting. This relationship is also important and impactful for the machine learning community, as we are on the quest of new deep learning technologies.

 * Application of the RDD method to spiking neural net training is novel. The reciprocal relationship of the causal effect to the synaptic strength is a very intuitive and elegant solution to the weight transport problem.

Cons:
 * From the reported results, it is not possible to decide whether RDD really outperforms Feedback Alignment (FA). The comparison is performed on only two data sets and each algorithm is better on one. Could the authors report results on at least two more data sets (however small or simple) during the rebuttal?

 * Fig and Table 1 report the same outcome. One of the two need to be removed.

Further Questions:
 * The Conv Net illustrated in Fig 2 panel A shares its weights with the biologically plausible net on panel B. Further, these two nets communicate for pre-training. How does the paper then isolate the contribution of the biologically plausible net to the prediction accuracy from the vanilla ConvNet? What would happen if we trained only the LIF net without a contact with the conv net?

 * Eq. 1 proposes induction of symmetry to solve the weight transform. At the extreme, this regularizer would make W and Y identical, boiling down to  a vanilla artificial neural net, which the ML community already knows wella nd performs with excellence. Would not having the biologically  implausible artificial neural model as the extreme solution contradict with the goal of biologically plausible learning? This would in the end make one conclude that the biological brain only performs a broken gradient descent.

Overall, this is a decent piece of work with some potential. My initial vote is a weak reject, as I  am at present missing sufficient evidence that the improved symmetry properties introduced by the causal inference scheme also brings an accuracy improvement over the vanilla feedback alignment method. I am open to improve to an accept if this evidence is provided and my aforementioned concerns primarily on the role of ConvNet are properly addressed during rebuttal.


--
Post-rebuttal: My only major concern was the lack of sufficient empirical evidence to support the idea. The updated version of the manuscript has properly addressed this issue by reporting results on additional data sets. The authors have also given enlightening clarifications to some of the open points I have raised earlier. Hence, I'm happy to increase my score.

**Experience Assessment:**

I have read many papers in this area.

**Review Assessment: Checking Correctness Of Derivations And Theory:**

I assessed the sensibility of the derivations and theory.

**Review Assessment: Checking Correctness Of Experiments:**

I assessed the sensibility of the experiments.

**Review Assessment: Thoroughness In Paper Reading:**

I made a quick assessment of this paper.

---

> ### Author Response · Authors · 2019-11-14
> **Response to Reviewer #2**
>
> "From the reported results, it is not possible to decide whether RDD really outperforms Feedback Alignment (FA). The comparison is performed on only two data sets and each algorithm is better on one."
>
> We are afraid that we must have been unclear in the original submission, so thank you for raising this. To clarify, RDD performs better than FA on all of the datasets we have investigated to date.
>
> "Could the authors report results on at least two more data sets (however small or simple) during the rebuttal?"
>
> This is a very valid request, and we are happy to oblige. We have no also tested on the SVHN and VOC datasets. RDD outperforms FA on both datasets (see updated Figure 5).
>
> "Fig and Table 1 report the same outcome. One of the two need to be removed."
>
> Fair point, we have removed Table 1, and provide both training and testing results in Figure 5 now.
>
> "The Conv Net illustrated in Fig 2 panel A shares its weights with the biologically plausible net on panel B. Further, these two nets communicate for pre-training. How does the paper then isolate the contribution of the biologically plausible net to the prediction accuracy from the vanilla ConvNet? What would happen if we trained only the LIF net without a contact with the conv net?"
>
> We now see that we were insufficiently clear in the original submission, so again, thank you for raising this. The interaction between the ConvNet and the LIF net is as follows: the two networks share weights, but the ConvNet is used for training the feedforward weights and measuring accuracy, while the LIF net is only for training the feedback weights. More specifically, on each epoch, we train the feedforward weights with the ConvNet, using the current setting of the feedback weights. This means that the transpose of the feedforward weights in the usual gradient update term is replaced with the current feedback weights. Then, we transfer the new feedforward weights from the ConvNet to the LIF net, and we train only the feedback weights. This continues: the feedback weights of the ConvNet are set to the new values from the LIF net, and so on. Thus, the LIF net is not learning to categorize the images, it is only learning the feedback weights, which get used by the ConvNet for the feedforward training. We have clarified this in the text and Figure 2A. We do this because our goal in this paper is simply to test the RDD algorithm's ability to learn good feedback weights, not to test the ability of an LIF net to perform categorization.
>
> "Eq. 1 proposes induction of symmetry to solve the weight transform. At the extreme, this regularizer would make W and Y identical, boiling down to  a vanilla artificial neural net, which the ML community already knows well and performs with excellence. Would not having the biologically  implausible artificial neural model as the extreme solution contradict with the goal of biologically plausible learning? This would in the end make one conclude that the biological brain only performs a broken gradient descent."
>
> The reviewer is correct that the symmetric alignment cost function would only be zero when perfect symmetry in weights is achieved. The reviewer is also correct that this would indicate that biological networks were approximating gradient descent. However, that is part of the point of this exercise. To date, no one has demonstrated how one can achieve efficient credit assignment in large networks without at least a good correlation with the true gradient. To be clear, we hypothesize that the brain may in fact have a means of estimating gradients, and that this would be achieved, in part, by ensuring symmetry between feedforward and feedback pathways. That may not be a "broken" gradient descent, in so far as there can be regularizing advantages to not always perfectly following the gradient. If the reviewer is interested in this perspective, they can read more in our recent review on the topic: Richards, et al. Nature Neuroscience 22, no. 11 (2019): 1761-1770.

---

### Official Review · AnonReviewer1 · 2019-10-23
**Official Blind Review #1**

**Rating:** 6

**Review:**

summary

This paper considers the "weight transport problem" which is the problem of ensuring that the feedforward weights $W_{ij}$ is the same as the feedback weights $W_{ji}$ in the spiking NN model of computation. This paper proposes a novel learning method for the feedback weights which depends on accurately estimating the causal effect of any spiking neuron on the other neurons deeper in the network. Additionally, they show that this method also minimizes a natural cost function. They run many experiments on FashionMNIST and CIFAR-10 to validate this and also show that for deeper networks this approaches the accuracy levels of GD-based algorithms.



comments

Overall I find this paper to be well-written and _accessible_ to someone who is not familiar with the biologically plausible learning algorithms. To overcome the massive computational burden, they employ a novel experimental setup. In particular, they use a separate non-spiking neural network to train the feedforward weights and use the spiking neurons only for alignment of weights. They have experimental evidence to show that this method is a legitimate workaround. I find their experimental setup and the results convincing to the best of my knowledge. The experimental results indeed show the claim that the proposed algorithm has the properties stated earlier (i.e., learns the feedback weights correctly and that using this to train deep neural nets provide better performance than weight alignment procedure). I must warn that I am not an expert in this area and thus, might miss some subtleties. Given this, it is also unclear to me why this problem is important and thus, would leave the judgement of this to other reviewers. Here I will score only based on the technical merit of the method used to solve the problem.

I had one minor comment on the arrangement of the writing of the paper. Section 4 starts off with "Results" but the earlier sub-sections are not really about the results. I would split section 4 as methodology/algorithm and include the everything until section 4.4. From sub section 4.5 onwards are the actual results.


overall decision

Without commenting on the importance of this problem, I think this paper merits an acceptance based on the technical content. The paper provides convincing experiments to test the properties the author claim the new algorithm has.

**Experience Assessment:**

I do not know much about this area.

**Review Assessment: Checking Correctness Of Derivations And Theory:**

I carefully checked the derivations and theory.

**Review Assessment: Checking Correctness Of Experiments:**

I assessed the sensibility of the experiments.

**Review Assessment: Thoroughness In Paper Reading:**

I read the paper thoroughly.

---

> ### Author Response · Authors · 2019-11-14
> **Response to Reviewer #1**
>
> "I had one minor comment on the arrangement of the writing of the paper. Section 4 starts off with "Results" but the earlier sub-sections are not really about the results. I would split section 4 as methodology/algorithm and include the everything until section 4.4. From sub section 4.5 onwards are the actual results."
>
> Yes, we see your point. We have split the materials into methods/results as requested.

---

### Official Review · AnonReviewer3 · 2019-10-23
**Official Blind Review #3**

**Rating:** 8

**Review:**

Strong paper in the direction of a more biologically plausible solution for the weight transport problem, where the forward and the backward weights need to be aligned. Earlier work for feedback alignment has included methods such as hard-coding sign symmetry. In this method, the authors show that a piece-wise linear model of the feedback as a function of the input given to a neuron can estimate the causal effect of a spike on downstream neurons. The authors propose a learning rule based on regression discontinuity design (RDD) and show that this leads to stronger alignment of weights (especially in earlier layers) compared to previous methods. The causal effect is measured directly from the discontinuity introduced while spiking - the difference between the outputs of the estimated piece-wise linear model at the point of discontinuity is used as the feedback.

Compared to feedback alignment, RDD-based pre-training demonstrates stronger alignment between forward and backward weights and better performance on CIFAR-10 and Fashion-MNIST datasets. Overall, the paper is very well written and addresses an important problem. The theoretical foundation, to my knowledge, is well studied.

**Experience Assessment:**

I do not know much about this area.

**Review Assessment: Checking Correctness Of Derivations And Theory:**

I assessed the sensibility of the derivations and theory.

**Review Assessment: Checking Correctness Of Experiments:**

I assessed the sensibility of the experiments.

**Review Assessment: Thoroughness In Paper Reading:**

I read the paper at least twice and used my best judgement in assessing the paper.

---

> ### Author Response · Authors · 2019-11-14
> **Response to Reviewer #3**
>
> Thank you for your comments!

---

### Author Response · Authors · 2019-11-14
**Response to Reviewers**

We would like to thank all of the reviewers for their encouraging comments and helpful critiques. We have updated the manuscript and believe that we have addressed the concerns that were raised. We provide responses to specific points below.

---

### Decision · Program_Chairs · 2019-12-19

**Decision:**

Accept (Poster)

**Comment:**

All authors agree the paper is well written, and there is a good consensus on acceptance.  The last reviewer was concerned about a lack of diversity in datasets, but this was addressed in the rebuttal.